# Position: When Incentives Backfire, Data Stops Being Human

**Sebastin Santy** [1]   **Prasanta Bhattacharya** [2]   **Manoel Horta Ribeiro** [3]   **Kelsey Allen** [4]   **Sewoong Oh** [1]

## Abstract

Progress in AI has relied on human-generated data, from annotator marketplaces to the wider Internet. However, the widespread use of large language models now threatens the quality and integrity of human-generated data on these very platforms. We argue that this issue goes beyond the immediate challenge of filtering AI-generated content – it reveals deeper flaws in how data collection systems are designed. Existing systems often prioritize speed, scale, and efficiency at the cost of intrinsic human motivation, leading to declining engagement and data quality. We propose that rethinking data collection systems to align with contributors' intrinsic motivations – rather than relying solely on external incentives – can help sustain high-quality data sourcing at scale while maintaining contributor trust and long-term participation.

## 1. Human Data in Crisis

Artificial Intelligence relies heavily on human-generated data to develop ever more capable models and systems that emulate human-like intelligent behavior. The primary sources of such data include: (1) human annotations from crowdsourcing platforms (e.g., Amazon MTurk), and (2) raw Internet data from communities like Wikipedia and Reddit. These two sources underpinned the last two major eras in AI: the deep learning era that began with AlexNet (Krizhevsky et al., 2012), powered by ImageNet (Deng et al., 2009) built via MTurk; and the pretrained language model era ushered in by BERT (Devlin et al., 2019) and enabled by large-scale Internet data.

This trajectory has reached an inflection point. With the

---

[1]University of Washington, USA [2]Institute of High Performance Computing (IHPC), Agency for Science, Technology and Research (A*STAR), 1 Fusionopolis Way, #16-16 Connexis, Singapore 138632, Republic of Singapore [3]Princeton University, USA [4]University of British Columbia. Correspondence to: Sebastin Santy <ssanty@cs.washington.edu>.

*Proceedings of the $42^{nd}$ International Conference on Machine Learning*, Vancouver, Canada. PMLR 267, 2025. Copyright 2025 by the author(s).

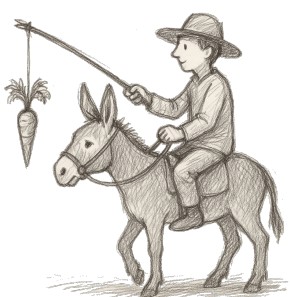

Figure 1. *Perpetual Donkey Machine.* It looks like the donkey could walk forever with the carrot just out of reach. But it won't, not forever. Reward a task the donkey would never do otherwise, and you get shortcuts – actions optimized only to reach the carrot. Reward a task it already does, and you risk erasing the inner drive that moved it in the first place – making it *less donkey*. Good incentives shape action. Flawed ones break the actor.

rise of generative language models like ChatGPT (OpenAI, 2023), the very sources of human data that fueled prior breakthroughs are getting destabilized. Contributors on annotation platforms are increasingly relying on LLMs to complete or expedite annotation tasks (Veselovsky et al., 2023; 2025), while the broader Internet is inundated with synthetic content (Brooks et al., 2024). As signals of authentic human behavior become harder to discern, the supply of high-quality data that was once the bedrock of progress in AI is at a risk of collapse. To compensate, much of ML research has started to lean on synthetic data – either to emulate human annotations (Dubois et al., 2024) or to mimic human behavior (Argyle et al., 2023; Park et al., 2022; 2023). However, these approaches have yet to reach the highest quality (Geng et al., 2024) and face significant challenges, such as model collapse (Taori & Hashimoto, 2023; Shumailov et al., 2024), keeping the ember of human-generated data still alive (Ashok & May, 2024).

We argue that the core issue is not new. Data collection platforms have long struggled with declining contributor engagement and quality. But the advent of LLMs has amplified these problems to the point where their continued viability has become uncertain (Pieces, 2025). At the heart of this crisis lies the question of human motivation: what drives people to contribute high-quality data in the first place?

## 2. Alternative Views

**Prevailing View.** A widely held assumption in machine learning is that high-quality human data can be reliably sourced through financial compensation. Crowdsourcing platforms operationalize this view, using task-based payments to drive participation and structure contributor behavior. While this strategy can guide contributors toward producing annotations, it often overlooks a critical factor: the contributors' intrinsic motivations. Studies have shown that over-reliance on extrinsic incentives not only risks diminishing intrinsic motives for engagement, but also erodes long-term performance on tasks. In our context, this would lead to a declining quality of data contributions.

**Position.** We argue that this incentive-centric view of human data collection is fundamentally limited. Relying solely on current compensation structures may ensure short-term throughput, but they fail to sustain the richness, authenticity, and *human-ness* of contributions over time. Instead, a more resilient approach must design for intrinsic motivation – supportive environments where participation is meaningful, voluntary, and rewarding in its own right. This does not preclude compensation; rather, it emphasizes that incentives must work with, not against, intrinsic motivation. This reframing is central to how future data systems should be built.

In this paper, we analyze the current data requirements in machine learning and how existing data collection systems attempt to meet them. We open up the black box of data collection – complex socio-technical systems shaped by human behavior, platform design, and technical constraints – drawing on foundational theories and experiments in the social sciences, particularly psychology and economics. In doing so, we examine the quantity-quality tradeoff and argue that, while this tradeoff may not be entirely eliminable, the overall quality and quantity of data can still be improved by identifying and removing factors that undermine intrinsic motivation. Finally, we contend that games offer a promising outlook, combining structure with sustained, voluntary participation in ways that promote long-term data quality and build trust.

## 3. Characterizing Human Data Needs

Progress in machine learning depends on data availability at a sufficient scale to inductively learn patterns from it (Kaplan et al., 2020; Hoffmann et al., 2022). This need for data has grown exponentially as learning algorithms have evolved from statistical to deep learning and pre-trained language models. The *quantity* of data has uncontestedly been the key consideration for the field (Halevy et al., 2009; Sutton, 2019), with any data source that adds several orders of magnitude to the size of existing datasets, such as data

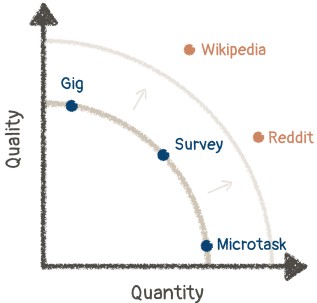

*Figure 2.* Illustration of a quantity-quality trade-off in data collection systems. Popular crowdwork platforms (e.g., MTurk / Microtask, Prolific / Survey, and UpWork / Gig) tend optimize for either scale or quality but struggle to achieve both at the same time. In contrast, data from sources not explicitly designed for collection, such as online collectives and communities (e.g., Wikipedia and Reddit), operate outside this trade-off, hinting at potential alternative paradigms.

crawled from the Internet, being considered indispensable. A general trend in machine learning regarding data sourcing, especially after the advent of pre-training with BERT (Devlin et al., 2019), has been to leverage sources of large data wherever they can be found, such as BookCorpus (Zhu et al., 2015), Wikipedia (Raffel et al., 2020), Reddit (Gokaslan & Cohen, 2019), and CommonCrawl (Common Crawl, 2021).

Recently, however, as datasets have grown larger, the importance of *quality* has become more apparent (Nguyen et al., 2022; Zhou et al., 2024; Lee et al., 2022). While learning algorithms have improved in extracting signal from noise, they still have limits when faced with excessive noise or irrelevant data (e.g., DataComp-LM discards 99% of data and Text-Image DataComp filters out 70%; Gadre et al. 2024; Li et al. 2024). Data quality has long mattered, but its significance has become clearer than ever as models trained on external proprietary datasets consistently outperform others on benchmarks and in real-world applications (Brown et al., 2020). This outperformance, often credited to the availability of "high-quality" proprietary datasets, such as paywalled content or licensed secondary sources (Bommasani et al., 2021), has pushed the data quality discourse to the forefront and is now a high priority in machine learning.

While both high quality and high quantity are critical for data sourcing, they often come at the expense of each other: improving one typically leads to a decline in the other. However, this trade-off is not necessarily intrinsic to the data itself, but a consequence of how systems are designed. We frame this trade-off as a Pareto frontier, as illustrated in Figure 2.

This framing helps clarify why data collection systems often struggle to balance quality and quantity. Platforms prioritizing quality, like freelance job platforms (e.g., UpWork), tend

to be slower with lower output, while high-throughput systems, like rapid crowdwork platforms (e.g., MTurk), scale efficiently but often sacrifice consistency and quality (Douglas et al., 2023). While this trade-off may never be fully eliminated in designed data collection systems, it is not a fixed constraint. Rather than removing it, the key is to expand the frontier by addressing inefficiencies in incentive design, annotation methods, and human oversight.

The dynamics of the quantity-quality trade-off are shaped by multiple interacting and, often, latent factors. Untangling these factors requires opening up current data collection systems and examining their trade-offs through the lens of human behavior, organizational processes, and technical constraints. At a system level, quality often depends on aligning intrinsic motivation with external incentives, while quantity is typically driven by process efficiency, often via task fragmentation and parallelization. As we explore later, these optimizations have unintended side effects, particularly when excessive fragmentation begins to erode participant engagement and long-term data quality. Understanding how these factors interact is key to rethinking data system design.

## 4. Understanding Data Quality

While quantity can be easily measured and is increasingly accessible through newer data sourcing methods, assessing quality has become increasingly challenging. As data availability has surged, what constitutes "high-quality" data for training machine learning models has become the subject of growing debate.

Defining data quality has long been a challenge in machine learning, as it lacks a universally applicable or quantifiable standard for what makes data "high-quality". Attempts to assess quality have either been subjective or objective in approach. Subjectively, quality is often linked to the trustworthiness of the source. For example, Wikipedia is often regarded as more reliable than data from personal blogs because a Wikipedia entry is deemed to have undergone some form of moderation (Albalak et al., 2024; Soldaini et al., 2024). Objectively, quality has been measured using statistical metrics, such as readability scores, or modeled metrics, such as GPT-3 Quality Filters (Gururangan et al., 2022) and DataComp's curated datasets (Li et al., 2024), which define quality in the context of their downstream use. Taken together, definitions either rely on perceived source credibility, measure intrinsic properties of the data, or evaluate quality based on how it performs in context – reflecting different assumptions about what quality means.

Existing definitions of data quality, whether based on credibility, features, or performance, are ultimately proxies. Quality is inherently situational: without clarity on intended use,

its meaning becomes elusive. In such cases, anchoring quality in *naturalness* – how people behave in routine contexts, online or offline, without interference – offers a more grounded perspective. Naturalness is harder to measure, since it depends not on post hoc proxies, but on observing the conditions under which data is generated. But for training models of human behavior or intelligence, the highest quality data may be that which reflects unprompted, incentive-free engagement with the world.

## 5. Human Factors in Quality

Crowdwork platforms used for data collection in machine learning (e.g., MTurk, Prolific, UpWork, ScaleAI) are designed with built-in compensation mechanisms. Rapid crowdsourcing platforms (e.g., MTurk) are commonly used for low-effort and low-pay tasks that can scale easily, but with unreliable quality (Douglas et al., 2023). In contrast, freelance job platforms (e.g., UpWork) tend to favor high-effort and higher-pay gigs, which require more deliberate and engaged participation, often leading to higher-quality outputs.

At first glance, this distinction aligns with a straightforward intuition: that financial compensation leads to greater effort and better quantity and/or quality contributions (e.g., Mason & Watts 2009; Ho et al. 2015; Shah & Zhou 2016; Laux et al. 2024). This assumption drives many current data collection practices, where compensation gradually turns into an incentive: a lever used by data collectors to improve data quality, or perceived as one by the contributors. What initially was a means to acknowledge value, starts getting instrumentalized – as if it were the primary determinant of high-quality engagement.

But financial compensation is not the only route to high-quality contributions. Some of the most valuable human-generated data comes from platforms where users are not financially compensated at all, such as Wikipedia, Reddit, and open-source communities. Here, participants contribute not because of pay, but because they find the activity meaningful, socially rewarding, or aligned with personal interests (Forte & Bruckman, 2005; Lampe et al., 2010). These platforms challenge the idea that quality data generation must solely depend on financial incentives, showing that intrinsic motivation alone can sustain long-term, high-quality engagement.

While extrinsic and intrinsic sources of motivation routinely coexist on digital platforms, their relationship is far from linear. Adding external incentives does not reliably enhance intrinsic motivation, and in some cases, it can undermine it. Similarly, removing incentives does not automatically restore intrinsic drive. The interplay between the two is complex, and understanding it is key to designing sustainable

data collection systems.

**Overjustification Effect** (Lepper et al., 1973) describes how external rewards can diminish intrinsic motivation and affect task performance. In a classic experiment, preschool children who already enjoyed drawing were divided into three groups: (1) those who were promised and received a reward, (2) those who received an unexpected reward, and (3) those who received no reward. This experiment revealed two key findings: first, children in the expected-reward group spent significantly less time drawing voluntarily after the reward was removed, compared to the other groups. Second, the drawings from the no-reward and unexpected-reward groups were rated as slightly higher in quality than those from the expected-reward group. These findings suggest that when an activity initially driven by intrinsic motivation is externally incentivized, removing rewards can lead to a decline in both engagement and performance.

**So why do external incentives backfire?** Two key theories help explain this phenomenon of motivational crowding-out: why extrinsic rewards can sometimes diminish intrinsic motivation to perform a task.

- **Self-Perception Theory (SPT)** (Bem, 1972) suggests that individuals infer their own attitudes and motivations by observing their past behaviors. When external rewards are introduced, they may begin to attribute their participation to the incentive rather than to their original or intrinsic interest. Over time, this shift in self-perception can make them less likely to continue the behavior once the reward is removed.
- **Self-Determination Theory (SDT)** (Deci, 1971; Deci et al., 2017) offers a broader framework by centering on autonomy, competence, and relatedness as key psychological needs for intrinsic motivation. When a task is externally controlled through incentives, individuals may feel a loss of autonomy, making the activity feel like an obligation rather than a choice. This helps explain why highly controlled environments often struggle to sustain long-term engagement.

Together, these insights highlight why relying solely on external rewards like financial incentives is not a sustainable driver for maintaining high-quality, long-term engagement.

**If intrinsic motivation is key to sustaining high-quality, long-term contributions, what role does external incentives play?** While excessive reliance on extrinsic rewards can be detrimental, insights from SDT and related theories suggest that their impact depends on purpose and design. In short, extrinsic rewards that conflict with individuals' psychological needs, such as autonomy, competence, and relatedness, are more likely to erode intrinsic motivation.

For example, when rewards or penalties are used to tightly regulate behavior, they can undermine a sense of autonomy and reduce motivation. By contrast, when external rewards are presented as *informational*, such as verbal recognition or an unexpected performance bonus, they can enhance a person's sense of competence and strengthen intrinsic motivation[1].

Moreover, well-designed incentives can serve as catalysts for behaviors that might not otherwise occur. Small, calibrated rewards can act as interventions – drawing attention to valuable behaviors without overwhelming intrinsic drive (Deci, 1971). Even in systems that favor intrinsically motivated behavior, such as laissez-faire environments where individuals act freely and bear the consequences, subtle incentive mechanisms can help align individual and collective goals, as in the case of *nudging* (Leonard, 2008).

The dynamic of compensation becoming incentive, and incentives prompting distorted behavior, if not carefully managed, is not unique to psychology – it also appears in economics, albeit through a different framing. Goodhart's Law suggests that when a measure (i.e., the value of something) becomes a target, it ceases to be a good measure – mirroring how the line between pay as compensation and its perception as an incentive can begin to blur. Perverse incentives, including the classic Cobra Effect, illustrate what happens when this blurred line is crossed: behavior begins to optimize for the incentive rather than the goal, often producing outcomes that actively undermine the original intent (Goodhart & Goodhart 1984; Kerr 1975; Siebert 2001).

**So, how do these social theories play out in real-world data ecosystems?** Consider the contrast between data collection platforms like MTurk and naturally occurring community platforms like Wikipedia. Social theories of motivation offer valuable insight into their divergent approaches to sustaining engagement.

Designed for control and throughput, crowdwork systems like MTurk end up relying on financial incentives to drive participation – the most immediate and measurable lever available. While intrinsic and extrinsic motivations may initially coexist, over time a crowding-out effect sets in. As intrinsic motivation erodes, systems compensate by tightening control and increasing financial rewards – triggering a vicious cycle where contributors prioritize efficiency over authenticity. This often results in gaming or shortcutting behavior, such as automating their annotation tasks using AI or other external tools, ultimately degrading data quality.

By contrast, community platforms like Wikipedia or Reddit depend primarily on intrinsic motivation. External rewards are minimal – badges, reputation systems, informal recognition – but even when present, they go beyond what's im-

---

[1]See Deci et al. (2017) for an excellent discussion on the different forms of extrinsic motivation and how they relate to performance and worker wellbeing

mediate or transactional, tapping into deeper psycho-social drivers like identity, belonging, and curiosity (Raacke & Bonds-Raacke, 2008; Ruggiero, 2000). Contributors show up because what they do feels meaningful to them.

This contrast reveals a crucial insight: building sustainable data systems is not just about offering better incentives – it requires designing environments that reinforce and protect participants' intrinsic motivation over time.

## 6. Human Factors in Quantity

Data collection systems differ not only in how they compensate contributors but also in the kinds of task structures they naturally support. At one end, rapid crowdwork platforms are well-suited to fragmenting work into micro-tasks (e.g., HITs on MTurk) that take seconds to a few minutes to complete, optimizing for speed and mass throughput (Malsburg, 2024). Survey-oriented platforms (e.g., Prolific) accommodate slightly longer, but still modular tasks, spanning minutes to hours (Prolific, 2024). On the other end, freelance job platforms (e.g., UpWork) structure work as longer-term projects, lasting days or weeks, and offering participants greater autonomy and depth of engagement (Upwork, 2024; at Home Smart, 2022).

Fragmenting work into repeatable micro-tasks enables parallelization across workers, replacing processes that would otherwise unfold serially. Beyond data collection, this reflects the nature of work itself: many tasks begin with uncertain goals, requiring creativity and deliberate effort. But to scale, they are often stripped down into more defined, repeatable steps. What starts as exploratory and thoughtful gradually becomes optimized for speed and efficiency.

Cognitively, this parallels the shift from System 2 processes – slow, effortful, and reflective – to System 1 processes, which are fast, automatic, and intuitive (Kahneman & Tversky, 2013). This transformation is not merely a natural evolution, but one that is actively accelerated by task fragmentation. An apt analogy is Fordism (Hounshell, 1984), which introduced the assembly line: a structured and repetitive workflow where modularized processes could be executed at scale.

However, as tasks become increasingly repetitive, fragmented, and controlled, contributors may grow estranged from the output of their labors (Braverman 1974, e.g., Glavin et al. 2021). The more modular and mechanical the work, the harder it becomes to find meaning or ownership in the end product. Over time, this detachment triggers a shift from fulfillment-driven to survival-oriented motivations, a regression in the hierarchy of needs (Maslow, 1943). For data collectors, this disengagement often results in declining data quality; for contributors, it can lead to diminished well-being (Gray & Suri, 2019).

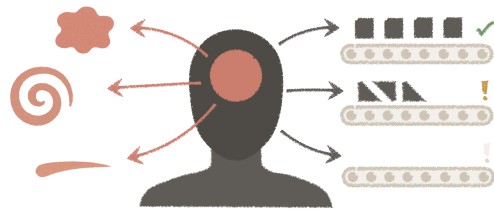

Internally Motivated          Externally Motivated

*Figure 3.* Intrinsic vs. Extrinsic Motivation: Internally motivated contributors are likely to produce human-like and diverse outputs, grounded in creativity and engagement. Externally motivated systems tend to favor controllability, structure and efficiency, often resulting in more uniform outputs that follow clearly defined goals. The figure illustrates how different motivational contexts can shape the nature and trajectory of human contributions.

Unlike physical labor, which benefits from built-in quality checks (e.g., material standards, inspections), knowledge-based tasks often lack such safeguards. In data annotation, for instance, there is often no immediate or reliable way to detect whether a task was completed thoughtfully or hastily (Klie et al., 2024a;b). As a result, quality can degrade quietly, with small errors compounding until the system becomes unsustainable.

**So, how does task fragmentation impact real-world data sourcing?** Micro-tasking on platforms like MTurk was once hailed as a transformative shift in computer science, enabling large-scale user studies (Bohannon, 2011; Kittur et al., 2013; Bernstein et al., 2011) and efficient data collection for machine learning (Deng et al., 2009). Over time, however, research has raised concerns about the reliance on "piece rate" or pay-per-task systems, favoring "quota" systems instead (Ikeda & Bernstein, 2016; Mason & Watts, 2009). These findings point to a degradation in task output quality when micro-tasking is pushed too far.

Beyond concerns about data quality, micro-tasking has also drawn sustained criticism for its effect on worker well-being. Recent works such as Ghost Work (Gray & Suri, 2019) and Anatomy of AI (Crawford & Joler, 2018) have illustrated the often invisible and exploitative nature of these atomized tasks. The non-physical nature of knowledge labor further exacerbates this issue, making the value of this work difficult to quantify (Martin et al., 2016). These dynamics grow more complex when microtasks are outsourced to countries with favorable exchange rates (Dicken, 2007), often to reduce costs – and in-turn, incentives – even further (Perrigo, 2023; Cheng, 2023), sometimes resulting in exploitative working conditions (Williams et al., 2022; Hao, 2022).

**What happens when tasks become so repetitive and unfulfilling that workers disengage from them entirely?** Over time, human-driven processes often shift from System

2 (deliberate & effortful) to System 1 (intuitive & fast). As tasks become more structured and predictable, they become prime targets for automation. In physical labor, this transition has been gradual with machines taking over repetitive, routine tasks, while humans focus on creative and uncertain work (Brynjolfsson & McAfee, 2014).

A similar shift is unfolding in knowledge-based work, where high-quality LLMs enable workers to offload mundane tasks, such as grammar corrections, spell-checks, and phrasing refinements, to AI. When used judiciously, this assistance promotes meaningful engagement and enhances productivity without compromising data quality. The problem arises when workers become over-reliant on LLMs, using them indiscriminately to complete entire tasks without much engagement or oversight (Veselovsky et al., 2023; 2025). Since knowledge-based tasks often lack clear-cut quality standards, it becomes harder to detect when quality slips, making it easier for disengaged or opportunistic behavior to go unchecked.

As a result, the transition to automation in data sourcing has been uneven and often chaotic. While repetitive physical labor was gradually and structurally offloaded to machines, knowledge work presents a more divided landscape – some advocate for fully replacing human contributors (e.g., Dubois et al. 2024), while others advocate for eliminating LLM usage entirely (e.g., Thorp 2023). However, fully relying on synthetic data risks model feedback loops and collapse (Taori & Hashimoto, 2023; Shumailov et al., 2024), while a complete ban might end up hurting human productivity and efficiency (Liao et al., 2024; Kreitmeir & Raschky, 2023). The most effective approach likely lies in between – where AI serves as a tool that productively and progressively supports human effort rather than a crutch for task completion (e.g., Ashok & May 2024; Qian et al. 2024).

In this evolving landscape, the role of intrinsic motivation becomes even more crucial. Workers must make deliberate choices on how to incorporate LLMs in ways that support rather than substitute meaningful engagement. Designing sustainable data collection systems is therefore not just about limiting LLM use for workers or maximizing automation with synthetic data – it is ultimately about creating an environment where contributors remain actively engaged with the task, rather than optimizing for speed at the cost of quality.

## 7. Expanding the Quality-Quantity Frontier

Inefficiencies in human factors and their resulting systemic designs limit how far data collection systems can push the quality–quantity frontier. External incentives, originally introduced to encourage participation, often end up hijacking intrinsic motivation over time. Likewise, task fragmenta-

tion, intended to simplify work and boost productivity, can spiral into microtasks so granular that contributors become disconnected from the broader purpose. Both become self-reinforcing vicious cycles that pull the frontier inward rather than pushing it outward.

In contrast, systems not *explicitly* designed for data collection – such as Wikipedia, Reddit, and open-source codebases – often yield high-quality, high-quantity data without deliberate optimization. Their success suggests that perhaps overfitting system design to data collection outcomes like quality and quantity may itself introduce inefficiencies, beyond just loss of naturalness or spontaneity. When these goals become explicit targets, designers often attempt to control them directly rather than allowing them to emerge from broader engagement dynamics. This impulse may stem from an illusion of control, where early performance gains reinforce the belief that increasingly fine-grained oversight will continue to improve outcomes, setting off a vicious cycle of continuing interventions, often reflective of cognitive entrenchment (Dane, 2010).

This highlights the **resolution of control** as a central design variable for addressing inefficiencies in data collection systems. While control can, in principle, be applied at any level of resolution, most systems tend to cluster around two regimes: environment-level and task-level control. Environment-level systems rely on contextual incentives (e.g., badges), social norms, and a shared purpose. Task-level systems, in contrast, depend on explicit task design and transactional incentives, such as monetary rewards. Both regimes often begin with a balance between intrinsic motivation and external incentives. In environment-level systems, the absence or weakening of external signals can lead to disengagement, as there are few levers to intervene directly when motivation fades. In task-level systems, declining intrinsic motivation often leads to heavier reliance on external (and often monetary) incentives, which can escalate and impact the sustainability of the system over time.

### 7.1. Rethinking Resolution of Control

Rather than managing individual microtasks, intentionally designed data collection systems may benefit from designing conditions that guide contributor engagement more holistically. Shifting from direct task management to a broader, environment-driven approach reduces the contributors' perception of controllability, which in turn reduces their tendency to attribute their actions to external rewards, helping preserve intrinsic motivation (Bem, 1972).

However, this shift presents a new challenge: relinquishing fine-grained task control requires designers to shape engagement at a more systemic level. Coarse-grained control, where engagement is shaped through platform design, incentive structures, and environmental cues, often takes

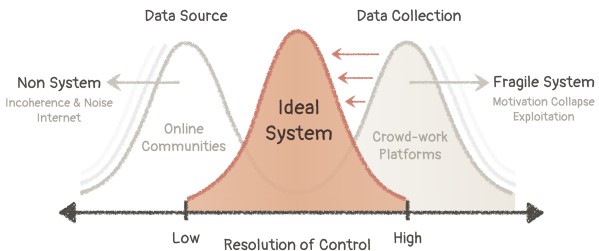

*Figure 4.* The Resolution-of-Control Spectrum. The horizontal axis orders human data pipelines by how tightly contributors' behavior is constrained, from environment-level, low control (left) to task-level, high control (right). Low-control systems include online knowledge bases and communities (e.g., Wikipedia, Reddit, GitHub) and high-control systems include crowd-work platforms (e.g., Upwork, Prolific, MTurk). Left unchecked, they drift toward disorder – the former towards incoherence, the latter towards motivational collapse. An ideal system sits at the center: a balanced, medium-resolution design that maximises the quality–quantity frontier by giving workers enough autonomy to stay motivated while providing sufficient control to obtain structured data.

longer to align with desired outcomes and requires greater up-front effort. Once in place, however, it can support more sustainable data collection, enabling both higher-quality and higher-quantity contributions, as seen in rare but influential examples.

### 7.2. Designing for the Middle

Product-integrated systems, such as deployed robots, offer concrete examples of mid-resolution control in practice. For example, robotic vacuum cleaners (e.g., Roomba) provide utility by cleaning homes, while simultaneously collecting spatial and navigation data to improve future performance (Astor, 2017). Users interact with the system for its core utility while passively generating high-quality data that can fuel further AI development (Brynjolfsson & McAfee, 2014). This model scales effectively, since data is collected continuously and passively through users' routine behavior, without requiring additional efforts to nudge contribution. This pattern extends to more complex, high-stakes systems like electric vehicles with driver-assist or autonomous driving features. Tesla, for instance, collects real-world driving data from its fleet to improve its self-driving AI systems (Karpathy, 2021; Tesla, 2021). These improvements not only enhance functionality for car owners but are also purported to drive future innovations, such as autonomous Robotaxis. Waymo similarly operates autonomous taxis in public settings, collecting large-scale data that has proven valuable for advancing computer vision research (Sun et al., 2020). What has recently been termed *the era of experience* reflects this shift toward systems that learn through human interactions while providing direct user value (Silver & Sutton, 2024).

However, replicating such large-scale, product-integrated data collection systems is often infeasible for intentionally designed data collection efforts. Product-integrated systems require significant hardware infrastructure, clearly articulated mutual benefits, and real-world applications supported by strong safeguards and privacy protections. For entities focused primarily on collecting human-generated data, building such ecosystems solely for data collection is neither feasible nor sustainable.

### 7.3. Trust & Alignment

Designing such systems ultimately hinges on building and sustaining trust. When engagement is not explicitly compensated or enforced, it must be sustained by a stable balance of expectations, incentives, and perceived fairness. These systems depend on informal social contracts, in which users participate not just for direct (financial) benefits, but because the broader arrangement feels legitimate and reciprocal.

This requirement is especially acute in data collection systems that aim to embed themselves in human experience. When contributors sense that their data is being used in ways that violate these expectations, such as serving third-party interests or commercial goals without consent, the relationship can quickly break down. This erosion of trust reflects principles described by Social Exchange Theory (Homans, 1958; Stafford & Kuiper, 2021) and Social Contract Theory (Kruikemeier et al., 2020; Fogel & Nehmad, 2009), which emphasize that cooperation and exchange depend on perceptions of reliability, fairness and mutual benefit.

The effects are already visible in creative and knowledge-sharing communities. Artists have recently protested against their work being scraped to train AI models without consent or compensation (Jiang et al., 2023)[2], with many calling for stronger protections against AI-generated art (Guardian, 2025). Similarly, Stack Overflow users, frustrated by the platform's shifting stance on AI use and its potential monetization of community contributions, have reportedly sought to alter or delete their posts as a form of protest or resistance against AI training (Technica, 2024; Hardware, 2024). These examples highlight the fragility of trust in data collection and the potential consequences when contributors feel that their data is being repurposed beyond its original intent.

## 8. Designing for the Middle: Games

Designing systems that integrate structured data collection with sustained, intrinsically motivated participation remains a central challenge. A key factor here is the resolution of control: too much control undermines intrinsic motivation and reduces participation to transactional compliance; too little results in incoherent, noisy, and ultimately low-value

---

[2]https://www.aitrainingstatement.org/

data. The most promising design space lies in the middle, where systems are intentionally structured yet rely on voluntary, self-directed engagement.

Games exemplify this balance. Players engage primarily for enjoyment, often fulfilling psychological needs such as competence and relatedness. At the same time, games are carefully designed with goals, rules, and constraints that elicit creativity, reasoning, and problem-solving (Koster, 2005). As a result, games generate rich, cognitively meaningful data that is useful for understanding decision-making and modeling intelligence (as already evidenced by their role in evaluating intelligence; Silver et al. 2016; Vinyals et al. 2019; Berner et al. 2019; FAIR). Games thus suggest a new frontier for AI data collection: one where structured environments driven by incentives and organic engagements driven by motivation can coexist by design.

### 8.1. Games for Data Collection

**Historical Precedent.** Games have long been explored as a tool for large-scale human annotation, most notably through von Ahn's Games with a Purpose (GWAP) (Von Ahn, 2006). Among the earliest and most influential examples was the *ESP Game* (Von Ahn & Dabbish, 2005), launched in 2004, which engaged thousands of players in a collaborative image-tagging game, producing millions of annotations. While players simply enjoyed the game, their interactions have been credited with helping bootstrap Google Image Search (Guardian, 2006), which previously relied on filenames of images, as large-scale labeled datasets like ImageNet were not available till much later, in 2009 (Deng et al., 2009).

Von Ahn argued that the billions of hours spent on games – such as the 9 billion hours spent playing Solitaire in 2003 alone, enough to build the Empire State Building in 6.8 hours or the Panama Canal in a day – could be repurposed for more meaningful tasks, inspiring the broader Games with a Purpose framework (Von Ahn, 2006). Other games in the series included *Peek-a-boom* (Von Ahn et al., 2006b), which collected image segmentation data, and *Verbosity* (Von Ahn et al., 2006a), aimed at gathering commonsense factual knowledge.

**Contemporary Efforts.** Recent efforts in machine learning have explored games for data collection, though few have reached the scale of earlier initiatives like GWAP. For instance, Google's *QuickDraw* (Ha & Eck, 2017) and AllenAI's *Iconary* (Clark et al., 2021) collect freehand drawings and pictographic communication. *Human or Not* (Jannai et al., 2023) gathers dialogue data through a gamified Turing Test, *Real or Fake Text* (Dugan et al., 2023) collects judgments on text authenticity, and *ArtWhisperer* (Vodrahalli & Zou, 2023) focuses on iterative prompt refinement for image generation. These games have collected data on

the order of tens to hundreds of thousands of interactions, demonstrating early promise.

However, building entirely new games tailored for data collection poses significant challenges. Designing engaging gameplay that simultaneously yields high-quality data clearly requires expertise beyond machine learning. Some projects circumvent this by leveraging existing games, where gameplay already sustains engagement. For example, Family Feud has been used for generating QA pairs (Boratko et al., 2020), and Minecraft as a collaborative environment for collecting dialogue data (Narayan-Chen et al., 2019).

In a similar vein, some efforts have introduced gamification elements, such as awarding points, stages of goal progression and completion, into traditionally non-game data collection tasks. For example, *CommonsenseQA 2.0* incentivizes users to craft questions that challenge AI models, while *Dynabench* adopts a competitive setup where humans try to "break" models by submitting failure cases, turning data collection into a game-like interaction loop (Talmor et al., 2022; Kiela et al., 2021).

### 8.2. Design Considerations

Designing games for data collection involves balancing two often competing goals: *(a) Optimizing data utility:* Ensuring that collected data serves AI/ML tasks – requiring structure, reliability, and task relevance. *(b) Preserving intrinsic motivation:* Crafting an experience that remains engaging over time, without artificial constraints or coercive incentives.

Striking this balance requires close collaboration between ML researchers and game designers to either (a) create new games purpose-built for data collection, (b) embed game-like mechanics into conventional data collection or annotation tasks, or (c) repurpose existing games that already capture natural engagement but require novel methods for extracting meaningful data. Each comes with trade-offs in resolution of control, scalability, and sustainability.

While past efforts have succeeded in optimizing for quality and quantity, sustaining trust remains challenging. For example, ReCAPTCHA, introduced in 2008 and used for annotating books and self-driving data (O'Malley, 2018; Anton, 2018), remains widely deployed today. However, its continued use has blurred the line between voluntary and coercive participation, with users expressing annoyance at the image-based challenges (Searles et al., 2023a;b), which can be seen as an early indication of eroding trust. Such cases underscore the difficulty of designing systems that simultaneously achieve all three: high-quality, high-quantity, and *high-trust* data collection – emphasizing trust as a third axis in a space traditionally optimized along the first two.

While no data collection games in machine learning have yet demonstrated sustained success, examples from other

domains suggest that long-term engagement and trust are achievable with thoughtful design. Scientific discovery platforms such as *Zooniverse*, which has engaged over a million volunteers in astronomy and other research (Cardamone et al., 2009; Lintott et al., 2008), *Lab in the Wild*, which supports large-scale behavioral studies in HCI and psychology (Reinecke & Gajos, 2015), and *FoldIt*, where players contribute to real protein folding problems (Khatib et al., 2011; Cooper et al., 2010), all demonstrate how sustained, motivated participation can be achieved outside traditional incentive structures. Even commercial games like *EVE Online* have integrated real-world scientific research tasks, such as through Project Discovery, while maintaining player engagement (LeBlanc, 2021). Research has also highlighted the potential of leveraging games for studying questions related to human cognition (Allen et al., 2024). Together, growing evidence shows that well-designed, game-like interfaces can yield high-quality data by attracting broad participation while preserving players' trust and motivations.

## 8.3. Trustworthy Design and Participation

As data collection moves into more immersive, naturalistic, and long-term contexts, the ethical stakes increase significantly. These systems cease to be merely transactional; they become embedded in everyday life, shaping behaviors, expectations, and even personal identities over time. Responsible design in such settings demands alignment with participants' values, rights, and expectations. Games, as familiar and culturally pervasive systems, offer a unique lens through which we can examine these emerging ethical and design considerations.

**Identifiable Data.** The use of human experiences as data for AI can be unsettling. Games, however, being distanced from real-world contexts, enable a clearer separation between real user data (e.g., identity or background) and gameplay data (e.g., actions, strategies, decisions), with the latter being of primary relevance to AI systems. This separation not only mitigates certain privacy concerns but also enables access to forms of behavioral data that are otherwise difficult to obtain. For example, access to natural human dialogue is often restricted by privacy constraints, and traditional data collection platforms struggle to collect rich, interactive exchanges. Synthetic datasets like SODA (Kim et al., 2023) aim to bridge the gap, but even the best LLM-generated data struggles to match the richness of human communication observed in games like Minecraft, where dialogue emerges authentically through goal-directed, context-rich interactions in pseudonymous environments (Narayan-Chen et al., 2019). Moreover, such environments offer an expansive representation of our physical world, and have served as valuable testbeds for multimodal and robotic AI tasks, including Habitat and AI2-THOR (Puig et al., 2023; Kolve et al., 2017). That said, games are not free from privacy risks, as real-world traces can occasionally surface in gameplay, highlighting the need to carefully distinguish user-identifiable behavior from in-game actions (Nair et al., 2022)

**Incentives and Manipulation.** Designing for motivation is not only about enabling participation – it is also about protecting it. Poorly calibrated incentives can unintentionally exploit psychological hooks, nudging players toward compulsive or performative behavior rather than authentic engagement. The growing concerns surrounding the use of lootboxes and microtransactions in games is a good example of this (Yokomitsu et al., 2021; Brady & Prentice, 2021). Furthermore, certain groups may disengage or be under-represented based on how the rewards are framed (Jun et al., 2017), leading to the resulting data being biased, which is problematic from a model training perspective.

**Rethinking Compensation Schemes.** Limiting pay as a lever does not reject compensation – quite the opposite. It clarifies its role: to acknowledge value. While games are often viewed as leisurely or "unproductive", repurposing them for data collection creates value, making it important to recognize and fairly compensate contributors. Unlike traditional annotation tasks, however, attribution in multiplayer, multisession games is complex, complicating the compensation process. One approach is to ensure contributors hold a stake in the value their data generates, potentially through decentralized models, e.g., (Oh et al., 2025). Regardless, compensation schemes must be carefully designed to avoid undermining intrinsic motivation. Replacing immediate and performance-based rewards with delayed or post-hoc recognition can be an alternative worth exploring.

## 9. Path Forward

In seeking sustainable approaches to human data collection for AI, we analyze existing data collection systems through the lens of the quantity-quality trade-off, arising from system design constraints that hinder simultaneous optimization of both. We discuss two specific factors affecting this trade-off: quality, shaped by external incentives and internal motivations; and quantity, driven by task fragmentation and efficiency. Drawing on decades of past work in the social sciences, we suggest that an over-reliance on external incentives and task control can gradually undermine intrinsic sources of motivation, leading to long-term declines in data quality. To mitigate this, we advocate a shift from controlling tasks to designing structured, trustworthy, adaptive, and engaging environments that can encourage sustained, meaningful, and self-directed participation. Games are a good example of such environments, where voluntary participation co-exists seamlessly with high-quality data generation. However, and importantly, envisioning such environments for data collection requires rethinking current incentive and compensation structures, along with questions of trust.

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
