# OpenReview forum: "Position: When Incentives Backfire, Data Stops Being Human"
_ICML.cc/2025/Position_Paper_Track — ICML 2025 Position Paper Track poster_

### Official Review · Reviewer_VBeJ · 2025-03-11

**Significance:** 2
**Argument Clarity:** 2
**Rating:** 2
**Confidence:** 3

**Questions:**

1. How can universal intrinsic incentive mechanisms be designed across diverse cultural contexts? For example, social recognition may drive participation in collectivist cultures, while self-actualization might dominate in individualist societies. Should system design be tailored accordingly?
2. The paper highlights the potential of gamified data collection, but existing examples (e.g., GWAP) remain small-scale. In the LLM era, how can gamified systems prevent users from exploiting AI-generated content?
3. Intrinsic motivation-driven systems may attract specific demographics (e.g., enthusiasts, professionals). Could this reduce data diversity? How can motivation and representativeness be balanced?

**Discussion Potential:**

3

**Paper Summary:**

This paper argues that the widespread adoption of generative large language models (LLMs) has led to a crisis in traditional human-generated data collection systems (e.g., MTurk), which rely heavily on extrinsic incentives (e.g., monetary rewards) and task fragmentation. The authors contend that these systems inherently undermine intrinsic motivation, resulting in declining data quality and long-term engagement. The paper advocates for rethinking data collection design by prioritizing intrinsic motivations (e.g., interest, social belonging) and explores gamified environments and structured platforms (e.g., Wikipedia, Reddit) as sustainable alternatives. Key contributions include: (1) a systematic analysis of the quality-quantity trade-off in data collection systems and its psychological underpinnings; (2) a framework for designing systems centered on intrinsic motivation; (3) a discussion of games as a promising frontier for scalable, high-quality data sourcing.

**Position:**

Yes

**Position In Title:**

Yes

**Related Work:**

2

**Strengths And Weaknesses:**

1.  It basically addresses the critical challenge of declining data quality in AI development, particularly the disruptive impact of LLMs on traditional data sources.
2. It integrates theories from psychology (Self-Determination Theory, Overjustification Effect), sociology (Labor Alienation), and economics (Incentive Theory) to substantiate claims.
Weaknesses:
1. While advocating for intrinsic motivation, the paper lacks concrete guidance on operationalizing or quantifying intrinsic drivers (e.g., universal incentive structures).
2. Limited discussion of practical barriers to gamified systems (e.g., development costs, user retention), potentially underestimating real-world complexity.
3. It fails to address how cultural differences (e.g., collectivist vs. individualist societies) might influence intrinsic motivation, risking oversimplification.
4.It relies heavily on existing literature and case studies without original experiments (e.g., direct comparisons of intrinsic/extrinsic incentives on data quality).
5. It may needs to analyze long-term sustainability of platforms like Zooniverse or Foldit to extract reusable design principles.
6. It may needs to explore technological tools (e.g., blockchain, decentralized platforms) to enhance trust in intrinsic motivation-driven systems.

**Support:**

2

---

> ### Author Rebuttal · Authors · 2025-04-01
>
> Thank you for your thoughtful review. We're glad that the paper’s focus on the challenges posed by declining data quality and its integration of interdisciplinary theory came through clearly. Below, we address the concerns you raised.
>
> **W1 & W2: Lack of guidance on operationalizing intrinsic motivation; limited discussion of practical barriers**
>
> We appreciate the reviewer’s thoughtful suggestions. Our primary aim in this position paper is to bring together insights from psychology, sociology, and ML to frame the opportunity space given the current challenge of obtaining authentic human data, not to provide a complete design blueprint. That said, we agree that more detail on how to operationalize intrinsic motivation, and a clearer accounting of practical constraints in gamified systems (e.g., retention, development cost), would strengthen the discussion.
>
> We will revise the text to include concrete factors that influence intrinsic motivation (e.g., autonomy, competence, relatedness) and discuss how these may manifest differently across contexts. As also noted by Reviewer av6X, drawing clearer connections between game elements and motivational theory can help ground the paper’s framing. To address this, we will expand the appendix with mappings between game mechanics and motivational theory.
>
> **W3: Failure to address cultural variation in intrinsic motivation; reliance on literature over original experiments**
>
> We have addressed how cultural variation influences intrinsic motivation on page 8, lines 398–410. We can certainly go into more depth; for example, collectivist cultures may value social recognition more strongly than individualist ones, where self-actualization or personal growth might dominate. However, going beyond a brief mention of these differences is not feasible within the provided length of this paper and may also risk digressing from our main arguments.
>
> Regarding the reliance on literature rather than original experiments, we note that conducting controlled experiments would require a very narrow scope, which would lose the essence of the broad position we are advocating. Our paper draws on several existing theories and experiments that have indicated these phenomena.
>
> **W4 & W5: Long-term sustainability and trust mechanisms in intrinsically motivated systems**
>
> These are great suggestions, and we’ve addressed related concerns in our response to Reviewer g2z8, particularly around what future systems might look like in terms of compensation and sustained engagement. As noted there, we’ve been exploring delayed or post-hoc recognition mechanisms that preserve intrinsic motivation while still acknowledging contributors’ value. We also agree that technologies like blockchain and decentralized systems could help build trust, attribution, and transparency, and we hint towards these possibilities in that section.
>
> As for long-term sustainability and reusable design principles, we agree that platforms like FoldIt and Zooniverse offer valuable case studies. However, extracting generalizable reusable design patterns from these systems is not straightforward, given their complexity and the interplay of social and technical factors. That said, we will make an effort to surface relevant patterns and highlight key takeaways that may inform future intrinsically motivated systems.
>
> --
>
> Thank you again for your review. We hope the revised version more clearly reflects the paper’s intent and opens new directions for both practical and theoretical work on sustainable data systems.

---

> > ### Comment · Reviewer_VBeJ · 2025-04-08
> >
> > I have gone through the rebuttal and other reviewers' comments. Now I am still not fully persuaded, especially w1 & w2. I don't think revise the text can really help resolve the related concerns. So I maintain the score.

---

> > > ### Author Response · Authors · 2025-04-09
> > >
> > > Thank you for your response and for checking other reviews. While we understand your position remains unchanged, we still wanted to acknowledge and address the explicit questions you raised, which we had inadvertently overlooked in our rebuttal, and now have the opportunity to respond to by replying to your comment:
> > >
> > > **1. How can universal intrinsic incentive mechanisms be designed across diverse cultural contexts?**
> > >
> > > This is a complex challenge, as your question rightly highlights. Intrinsic motivations are not universal; they are layered across different group identities that people associate with -- including cultural, social, and personal levels. Still, at their core, they reflect shared psychological needs, as Ryan and Deci state in a later paper:
> > >
> > > "The very fact that need satisfaction is facilitated by the internalization and integration of culturally endorsed values and behaviors suggests that individuals are likely to express their competence, autonomy, and relatedness differently within cultures that hold different values."
> > >
> > > There is no single intrinsic incentive mechanism that works universally across individuals or cultural contexts. However, cross-cultural research in psychology (e.g., the Inglehart–Welzel map, Hofstede's or Schwartz's dimensions), and work in organizational behavior (e.g., Erin Meyer's The Culture Map) provide rich insights into how these universal needs are culturally expressed which can inform how we design our systems. For example, low-context cultures (e.g., Germany, US) may respond more strongly to incentives emphasizing individual mastery or competition, while high-context cultures (e.g., Japan, South-East Asia) may be more motivated by collective recognition or social harmony.
> > >
> > > We see this not as a limitation, but as an opportunity. If we can design intrinsically motivated data collection systems across cultural contexts (e.g., Wikipedia), we don't just improve data quality, but also take a meaningful step toward building ML systems that reflect and serve a truly global population, as opposed to Western-centricity in ML today [1,2].
> > >
> > > [1] Ramaswamy, V. Geode: A geographically diverse evaluation dataset for object recognition. NeurIPS 2023.
> > >
> > > [2] Nguyen, T. Multilingual diversity improves vision-language representations. NeurIPS 2024.
> > >
> > > **2. How can gamified systems in the LLM era prevent AI-generated content from polluting human data collection?**
> > >
> > > We start from a simple observation: people don't play games because they are paid -- in fact, many pay to play them. This suggests that games tap into something fundamentally human: agency, challenge, curiosity. When someone is intrinsically motivated, they are less interested in shortcuts and more invested in meaningful engagement.
> > >
> > > This is often reflected in gaming communities, where players reject the option to unlock all levels upfront, even when it's free, choosing instead to progress through each level themselves [3,4]. This extends to hobbies, where shortcuts are available but often ignored because the activity itself is the reward.
> > >
> > > We believe that if task design is able to tap into contributors' intrinsic motivations, they will be less likely to use LLMs as a crutch. While this may not prevent LLM misuse entirely, it encourages deeper engagement and reduces the temptation to bypass the activity just to reach an outcome.
> > >
> > > [3] Reddit posts from players:
> > >
> > > https://www.reddit.com/r/truegaming/comments/4tw092/would_you_skip_content_in_a_game_why/
> > >
> > > https://www.reddit.com/r/truegaming/comments/kuyboq/guides_kinda_ruin_some_games_yeah/
> > >
> > > https://www.reddit.com/r/truegaming/comments/33sfil/do_you_feel_like_achievements_have_ruined_the_way/
> > >
> > > [4] This Psychological Trick Makes Rewards Backfire. https://www.youtube.com/watch?v=1ypOUn6rThM
> > >
> > >
> > > **3. Could reliance on intrinsically motivated contributors reduce diversity?**
> > >
> > > Diversity can be a challenge in both intrinsically and extrinsically motivated systems. Beyond cultural skew (e.g., MTurk workers predominantly from US and India), purely monetary may sample only financially constrained workers willing to accept low pay. For example, Bigham [5] describes how, after attempting a half-day shift on MTurk, they were able to earn only &#36; 8.50 (around 2.13 per hour) despite actively searching for and completing tasks. Intrinsically motivated systems have sampling problems, but on a different axis due to variation in what motivates different people but is studied heavily in research and business.
> > >
> > > Motivation and representativeness are not mutually exclusive, as shown by data sources like Wikipedia, built by broad, globally distributed communities rather than a narrow professional class -- though participation patterns still vary across regions [6].
> > >
> > > [5] Bigham. My MTurk (half) Workday. https://www.cs.cmu.edu/~jbigham/posts/2014/half-workday-as-turker.html
> > >
> > > [6] Hwang, S., & Shaw, A. Rules and Rule-Making in the Five Largest Wikipedias. ICWSM 2022. (thanks to reviewer g2z8 for the pointer)

---

### Official Review · Reviewer_g2z8 · 2025-03-11

**Significance:** 4
**Argument Clarity:** 2
**Rating:** 3
**Confidence:** 4

**Questions:**

Drawing mainly on the points from above, but framing as clean questions:
- Is it critical that this proposal be coupled with some kind of economic intervention?
- What specific steps should industry organizations take vs. Academics?
- To what extent would widespread shift towards intrinsic data collection create new tensions with privacy?

**Discussion Potential:**

4

**Paper Summary:**

Summary: This paper argues that research in AI, and AI-operating organizations, should design data collection pipelines that aim to align with intrinsic motivations people might have to create data. To define “intrinsic” data, the  paper highlights the contrast between MTurk style data collection and Wikipedia style data collection, and motivates the need to shift towards intrinsic motivation to contend with the use of LLMs in the generation of future data. Ultimately, the position of the paper is that intrinsic motivations will allow for higher quality data, so organizations have a direct incentive themselves to focus on this set of incentives.

**Position:**

Yes

**Position In Title:**

Yes

**Related Work:**

2

**Strengths And Weaknesses:**

## Strengths

- Position is generally clear and has direct actionable changes for the key audiences and their contexts (academic research in ML/AI, industry practitioners who are thinking about where to get their next dataset from).
- The idea of using naturalness for data quality estimation is interesting, and this position paper could help spark further technical work in this space.
- The discussion of various sociological and economic theories and shifting more emphasis in ML/AI towards data production more generally could have positive impact.
- Coverage of psychological explanations for how people respond to incentives could be useful for people in the ICML community who are interested in data collection pipeline design.

## Weaknesses
- While there is a citation to Kittur et al 2013 and related work, the overall engagement with social computing and HCI research that has studied data creation, crowdsourcing, human computation, peer production, etc. is very light.
    - In particular, it might make sense to pull in some findings or framings from work on Wikipedia governance, issues with online communities (e.g., why people may not enjoy using StackOverflow), work that has directly discussed the relationship between such data and AI, etc.
        - As a few examples that may be helpful (though not to imply these are “missing”, but just that these perspective may help strengthen): https://ojs.aaai.org/index.php/ICWSM/article/view/19297, https://doi.org/10.2753/MIS0742-1222260103, https://link.springer.com/article/10.1007/s10664-019-09685-x, https://dl.acm.org/doi/abs/10.1145/3593013.3594070
    - Importantly, I think highlighting the complex role of organizations and social structures in the success stories is important (and distinguishes Wikipedia quite heavily from GWAPs)
- I’m not sure the “alternate view” section in the current draft captures all the dynamics of the current landscape. Arguably, intrinsically motivated data has already had massive impact on AI research, as evidenced by the dominance of e.g. Wikipedia as a data source. It seems from the perspective of an academic or an AI company, moving towards intrinsic data is already a natural choice just from a cost-savings perspective. And while firms/labs are certainly continuing to hire contractors and explore the use of organizations like Scale AI, I think the notion that the majority stance is characterized by “depending on and modulating external incentives” could be clarified. So there are still-present forces that will push for intrinsic data (and in some ways that create tension with policy, data markets, etc.)
    - Concretely, in the context of a position paper and the requirements for this track, this section would heavily benefit from more citations and even direct quotations to clarify the exact perspective being argued against.

While not a weakness, one tension in the current draft is that arguments are made against micro-tasking on the basis of past work that has highlighted low pay, poor work conditions, etc. However, the current draft doesn’t discuss anything along the lines of how people might be paid going forward. Rather, it perhaps suggests a future world in which more people have better jobs with free time to e.g. contribute to Wikipedia or play new LLM-era GWAPs, but it would be nice to spell this claim/vision out.

**Support:**

2

---

> ### Author Rebuttal · Authors · 2025-04-01
>
> Thank you for your detailed and thoughtful review. We’re glad that the paper’s core contributions came through clearly, including its relevance for different audiences, the potential of naturalness as a framing for data quality, and its integration of sociological, economic, and psychological perspectives. Below, we address the points you raised.
>
> **W1: Limited engagement with HCI and social computing literature**
>
> We appreciate this point. Our goal with this paper is to initiate a discussion in machine learning about the role of intrinsic motivation in data collection, for which we draw from a broad span of research in psychology, sociology, economics, HCI, and machine learning. Connecting the threads across so many fields and their prominent theories meant that engaging in depth with any one area or topic risked digressing from the main points we intended to make and the conceptual footing we hoped to offer ML researchers, rather than serving as a comprehensive primer.
>
> That said, the points raised are important and we appreciate the specific references. We plan to incorporate them where relevant, particularly in sections that discuss Wikipedia and StackOverflow. These works offer helpful context on how community norms, governance structures, and contributor experiences shape participation, which directly complements our discussion of motivation and long-term sustainability.
>
> **W2: The “alternate view” section oversimplifies the current landscape**
>
> We agree that the "Alternate View" section would benefit from a more careful articulation of what the current predominant stance actually is. It is true that intrinsically motivated data sources such as Wikipedia and Reddit have had a major impact on AI research already. Our intention was not to deny this impact, but rather to point out that this reliance has largely been incidental and under-theorized in ML research, which continues to build new data pipelines around extrinsic incentive structures. We will revise alternate views to make the “intentional” vs.”incidental” part more explicit in addition to alternate view just being about external incentives.
>
> To clarify what we mean by the majority stance of “depending on and modulating external incentives”: we are primarily referring to the common practice where, when something feels off in the quality of data or engagement of contributors, the first lever that data collectors reach for is often pricing per task. Our suggestion is to also begin considering intrinsic motivational levers, which may lead to similar outcomes in a more sustainable way, even if they require more effort upfront.
>
> We appreciate the reviewer’s suggestion and agree that a clearer articulation of the majoritarian stance can help make the assumptions more explicit and create a sharper contrast with the direction we propose, ultimately strengthening the paper’s core argument.
>
> **W3: Unclear vision for compensation or labor structure in future systems**
>
> This is a brilliant point and one that we have thought more deeply about since the original submission. While the paper critiques micro-tasking for its limitations, our intention is not to suggest a future where labor is uncompensated. In our revised (personal) manuscript, we had already incorporated our thoughts on this, and we quote the relevant text here:
>
> “While games are often seen as leisure activities or ‘unproductive’, repurposing them for data collection contributes to value creation, in which case, it is important to recognize and fairly compensate contributors. The challenge is that, unlike traditional annotation tasks, AI data attribution is ambiguous, making it difficult to measure and distribute compensation effectively. One way to address this is by ensuring that contributors have a stake in the value their data generates, potentially through decentralized blockchain-like compensation models. However, in general, compensation must be carefully structured to avoid distorting intrinsic motivation. If participants anticipate compensation upfront, their engagement risks becoming extrinsically driven, leading to lower-quality contributions. A potential solution is delayed, post-hoc recognition, where participants are acknowledged after the fact, ensuring they remain intrinsically engaged while still benefiting from their contributions.”
>
> We will make sure this vision is clearly articulated in the revised version, as it ties directly into concerns around labor, fairness, and long-term sustainability of data collection efforts.
>
>
> --
>
> Thank you again for your constructive feedback. We hope the revised version more clearly communicates the paper’s intent and helps ground this conversation in the broader ecosystem

---

> > ### Comment · Reviewer_g2z8 · 2025-04-02
> >
> > Thanks to the authors for these comments and the responses to other reviewers. Taken together, I think this suggests many of the opportunities for improvement laid out across reviews are very actionable in minor revisions.
> >
> > - The clarifications on W1 is reasonable
> > - The clarifications on W2 is very helpful
> > - The clarifications on W3 are probably the most "high risk"  / hardest, but this approach seems reasonable in connecting the gamification approach to broader economic questions.

---

> > > ### Author Response · Authors · 2025-04-08
> > >
> > > Thank you for your follow-up and for engaging with both our response and those of other reviewers. We're glad to hear that the clarifications were helpful. After submitting our rebuttal, we realized we had inadvertently missed answering explicit questions raised across reviews, including yours. Since we now have the opportunity to respond to your comment, we'd like to take a moment to address your questions directly:
> > >
> > > **1. Is it critical that this proposal be coupled with some kind of economic intervention?**
> > >
> > > We did not initially think it was necessary to explicitly address economic interventions in the original submission, but reflecting on the reviews, we now realize this is important, particularly given the risk of misinterpreting our argument as suggesting that intrinsic motivation and external rewards are opposites (a concern raised by Reviewer BHDJ). We are not opposing monetary incentives, as might be inferred from our critique of overreliance on them, but are emphasizing the need to strengthen intrinsic motivation, without which we risk a vicious cycle. You are right that bringing intrinsic motivation into the mix requires rethinking the economic models, and we will bring this argument more to the forefront of the paper.
> > >
> > > As mentioned in our response to you, one model we've considered is post-hoc compensation or recognition, where contributors are rewarded retroactively if their data proves valuable. This allows intrinsic motivation to drive engagement while still enabling compensation for work done. We believe this kind of delayed or non-prescriptive reward structure is worth exploring further.
> > >
> > > **2. What specific steps should industry organizations take vs. academics?**
> > >
> > > This is a great question. We believe the structural differences between industry and academia are actually beneficial for this direction: industry tends to operate on clear monetary exchanges and can continue to push on that front. But this is also a timely opportunity for academia to bring intrinsic motivation back into focus -- after all, researchers know firsthand the value of working on problems for their own sake.
> > >
> > > Industry is already leading the charge with platforms like MTurk, Prolific, and ScaleAI, and is well-positioned to experiment with large-scale incentive infrastructures. Alternatively, for non–data-collection industries, there is also potential to embed these systems into platforms where users already spend time. Academia, on the other hand, can take on more foundational work: exploring motivational framings, running controlled studies, and prototyping systems that may not yet be viable at scale. We see this as a complementary relationship, where academia helps de-risk ideas that industry can refine and scale.
> > >
> > > This complementarity has been evident in AI and ML research, with academic research leading to large-scale impact through industry.  However, we do not see foundational work focused on ML data collection itself, perhaps because it meddles with areas outside CS, like psychology, and combining them can feel methodologically or institutionally messy.  At the same time, the need for higher-quality data is only growing, and we have seen many efforts to improve data quality on technical fronts, albeit most end up treating the symptoms rather than the source.
> > >
> > > We also get the sense that many academic researchers feel constrained or outmatched when it comes to shaping data infrastructure, given the scale of industry pipelines. But we believe this is exactly the kind of long-horizon, interdisciplinary challenge that academic research is well-positioned to take on. Our hope is that this paper helps signal both the importance and the opportunity of doing so.
> > >
> > > **3. To what extent would a widespread shift towards intrinsic data collection create new tensions with privacy?**
> > >
> > > This is an important point, and like your compensation question, it's one we've reflected on since the original submission. Here is a quote from our manuscript:
> > >
> > > "Beyond incentive design, another fundamental challenge is ensuring that game-derived data remains both ethical and useful. Not all games inherently separate real-world contexts, which may necessitate distinguishing user-identifiable behavior from game-playing behavior for privacy reasons [Nair et al. 2022]."
> > >
> > > Privacy concerns will likely vary depending on context, especially in systems built around intrinsic motivation, where interactions are often more natural and unstructured. Some risks, such as behavioral traceability or exposure of personal traits, may only surface in real-world deployment. We see this as an area where careful system design and consent processes will be crucial.
> > >
> > > We are continuing to consider these questions and would welcome any frameworks or directions you think may be especially relevant.
> > >
> > > [Nair et al. 2022] Vivek Nair, Gonzalo Munilla Garrido, and Dawn Song. 2022. Exploring the unprecedented privacy risks of the metaverse. arXiv:2207.13176

---

### Official Review · Reviewer_BHDJ · 2025-03-13

**Significance:** 2
**Argument Clarity:** 3
**Rating:** 2
**Confidence:** 4

**Questions:**

What empirical support do you have for your position?
Can monetary compensation ever approach or exceed inartistic motivation?
Is gamification scalable in the same way as monetary compensation?
Are there ethical concerns with gathering data without monetary compensation?

**Discussion Potential:**

2

**Paper Summary:**

This paper argues that AI needs new strategies for collecting real human-generated data—especially as large language models (LLMs) make it easy to produce synthetic text that dilutes authentic contributions. Traditional data-collection platforms (like MTurk) center on rapid, low-cost crowdwork and often rely on pay-per-task incentives. However, merely increasing external rewards tends to erode intrinsic motivation, causing data quality to deteriorate over time. The paper proposes shifting from tightly controlled, fragmented micro-tasks to environments where people are intrinsically motivated—such as Wikipedia-style collaborative communities or well-designed games with a purpose—so that both the quantity and the naturalness (quality) of collected data improve. Ultimately, the authors suggest that AI progress depends on fostering genuine human engagement and trust rather than simply ramping up short-term incentives.

**Position:**

Yes

**Position In Title:**

Yes

**Related Work:**

3

**Strengths And Weaknesses:**

Strengths:

The paper spotlights a real and urgent issue: High quality date is needed to sustain AI development and current incentives may not be well aligned.

By connecting data quality to intrinsic motivation, the paper presents a nuanced analysis of why simply raising crowdworker pay or implementing stricter controls may backfire.

The authors propose that games as a way to strike a balance between structure and fun while capturing large amounts of natural human behavior. This is an interesting and creative proposal

Weaknesses:

The authors acknowledge that increasing pay is an intuitive solution to the problems they address. They claim that some of the highest quality human contributions come from Wikipedia and Reddit. However, it seems that the highest quality human contributions are generally made for financial rewards: highly paid lawyers likely produce higher quality analyses than those found on legal subreddits. In absence of a fuller comparison of work product produced for high compensation to content created due to intrinsic motivations, it seems that the authors argument is weak and that the challenge they flag could be addressed more effectively by simply paying far more for data.

The authors also fail to address the ethical concerns central to the idea of using psychology rather than monetary compensation to gather data that, at least in some cases, has significant commercial value.

**Support:**

3

---

> ### Author Rebuttal · Authors · 2025-04-01
>
> Thank you for your review and for highlighting key strengths of the paper, including the urgency of improving data quality, the nuance around how incentives interact with motivation, and the creative potential of games as data collection systems. Below, we address the concerns you raised.
>
> **W1: Financial compensation might still be more effective than intrinsic motivation**
>
> We appreciate the reviewer raising this important tension. We want to clarify that our position is not that financial incentives cannot produce high-quality work – clearly they can, especially in high-skill domains like law or medicine. Our argument is that over-reliance on only extrinsic rewards can diminish intrinsic motivation, particularly in domains that lack skill differentiation, quality control mechanisms, and moral and legal accountability. Unlike professions like law and medicine, large-scale data collection environments do not have such inherent structures that uphold quality or ethos, making them fertile ground for the vicious cycle of over-reliance on external incentives.
>
> We also do not present intrinsic and extrinsic motivations as opposites, but rather as different levers that can be thoughtfully combined. Both are valuable, and balance is ideal. However, extrinsic incentives are often over-relied upon due to their immediate convenience, which then becomes the default, introducing a number of downstream issues we explore in the paper.
>
> Thanks again for raising this point – we will clarify this distinction in the paper, as it could be a common misinterpretation of our framing.
>
> **W2: Ethical concerns around using intrinsic motivation to gather commercially valuable data**
>
> We thank the reviewer for raising ethical concerns around systems that rely on intrinsic motivation. This is a crucial dimension, especially as data collected through such systems may be used in commercial contexts. Our paper already discusses the risk of undermining contributor trust (Section 7), particularly when contributions are repurposed without consent or reciprocity.
>
> That said, we respectfully note that all incentive structures – including monetary ones – are shaped by psychological dynamics. While monetary compensation provides a clearer contractual exchange, it also introduces new power asymmetries (e.g., low pay for high-value data, exploitative global labor markets), and can still lead to contributors feeling manipulated or undervalued. In our view, the ethics of data collection hinge less on whether compensation is monetary or intrinsic, and more on issues of transparency, consent, and fair value exchange.
>
> We will expand our discussion of labor and fairness, especially by drawing on policy perspectives from intellectual property and labor law. We appreciate the reviewer’s push to bring these concerns more fully into focus.
>
> --
>
> Thank you again for your engagement. We hope the revisions help clarify that this paper is not a rejection of financial incentives, but a call to design data systems that are more aligned, ethical, and sustainable.

---

### Official Review · Reviewer_av6X · 2025-03-13

**Significance:** 4
**Argument Clarity:** 2
**Rating:** 3
**Confidence:** 3

**Questions:**

- Can you think of a more unified way to think about the proposed games approach such that it encapsulates the mentioned related work?
- Are there any potential risks or downsides to consider with games?

**Discussion Potential:**

3

**Paper Summary:**

- This position paper argues for rethinking how high-quality and quantity of human data is collected for AI/ML training.
- In particular, the paper discusses aligning collection efforts with external and internal human motivation to ensure quality and fragmenting tasks appropriately to ensure quantity.
- The paper proposes the use of games as a collection mechanism that may lie on the Pareto frontier of quantity and quality.

## update after rebuttal
The authors' proposed edits in response to my concerns seem reasonable. I will maintain my score.

**Position:**

Yes

**Position In Title:**

Yes

**Related Work:**

4

**Strengths And Weaknesses:**

Strengths:
- This position paper lays out the challenges of incentivizing high quality and quantity of data collection both from the data collection perspective and from the human perspective.
- This topic is certainly an important and relevant one as researchers continue to collect human data for both training and evaluation efforts.

Weaknesses:
- As written, the position paper pivots to games as a solution for incentivizing quality data, which seems to come out of nowhere at the end of the paper. In contrast, this would ideally be built into the core argument of the work.
- he paper cites many example works that incorporate gamification elements. However, this section could be significantly strengthened if the authors could more clearly lay out game elements that align with the incentivization discussion from earlier in the paper, show how existing efforts satisfy various elements.

**Support:**

3

---

> ### Author Rebuttal · Authors · 2025-04-01
>
> Thank you for your thoughtful review and for highlighting the significance of the problem space. We’re glad the framing of data quality and incentive challenges came through clearly. Below we address the concerns you raised.
>
> **W1: Games appear abruptly in the paper and feel disconnected from the core argument**
>
> We appreciate this observation. We intended to decouple the theoretical framing (incentives, psychology, data quality) that examines the problems with current human data collection systems from potential implementations like games. However, this structure may have unintentionally made the games section feel like an add-on.
>
> As a side note, we do not propose games as the only solution. They represent one promising design space, particularly because they allow for low-stakes experimentation, are typically built around engagement, and span a broad spectrum that can support the collection of diverse reasoning data (from logical planning to social reasoning). Software products can also enable similar data collection, such as Duolingo’s early goal of translating the web while users learn languages [1], or Anthropic’s use of user interaction data to understand markets [2]. However, these approaches require product-market fit, operate in high-stakes domains, and rarely offer the same flexibility or control over what data is collected and how.
>
> Coming back, this is why we hesitate to tightly interweave the theoretical framing with the games section. Instead, we will clarify this more explicitly when introducing the games section, and revise it to make the transition feel more fluid – but we are open to suggestions otherwise. In particular, we will show how common game mechanics (such as competition, personal challenge, narrative, and social belonging) directly map to the motivational frameworks discussed earlier, such as Self-Determination Theory and Self-Perception Theory, as we elaborate further in response to your next point.
>
> [1] Making the web accessible by translating the web through people learning new languages (concept of Duolingo): https://www.youtube.com/watch?v=i7QftS-yrL0
>
> [2] Anthropic's Economic Index working with user data collected from their chat models: https://www.anthropic.com/news/the-anthropic-economic-index
>
>
> **W2: Mapping game elements to motivational theory could be clearer**
>
> Extending our previous discussion, we agree that drawing clearer connections between game mechanics and psychological theories of motivation would strengthen the paper. Our intent with this paper was to first build a shared vocabulary across ML, psychology, and HCI, and then introduce games as one design space to explore. We avoided going too deep in the main text to avoid overwhelming the core argument, but we’re happy to expand this in an appendix.
>
> For example, we can include illustrative alignments such as:
> - Autonomy: open-ended sandbox games e.g., Minecraft
> - Competence: puzzle or strategy games with progressive skill building e.g. Tetris, RTS games
> - Relatedness: multiplayer and community-based platforms e.g. FoldIt, Zooniverse, Among Us
>
> In addition to this, foundational game design works such as *A Theory of Fun* (Koster, 2005) and *Rules of Play* (Salen & Zimmerman, 2003) offer systematic frameworks for understanding play, engagement, and motivation. We believe adding these references can provide useful starting points for ML researchers interested in exploring this direction more deeply.
>
> --
>
> Overall, we appreciate your constructive feedback on better integrating the games section with the earlier theoretical framing. We hope the revised version more clearly unifies these threads and communicates the design opportunities that lie ahead.

---

### Official Review · Reviewer_pRRi · 2025-03-16

**Significance:** 4
**Argument Clarity:** 3
**Rating:** 2
**Confidence:** 4

**Questions:**

1. What are some specific case studies or examples that demonstrate successful implementations of gamified data collection systems? What lessons can be drawn from these examples?

2. How can data collection systems be designed to accommodate the different intrinsic motivations of different cultures or demographic groups?

**Discussion Potential:**

3

**Paper Summary:**

The submission proposes a position that focuses on sustaining high-quality human-generated data in machine learning. Specifically, traditional data collection systems often prioritize speed and efficiency at the expense of data quality and contributor engagement. And the increasing reliance on AI-generated content has made it harder to obtain authentic human data, raising concerns about a potential shortage. In this position, data collection systems should align them with contributors' intrinsic motivations rather than relying solely on external incentives, as enhancing intrinsic motivation can lead to greater engagement and improved data quality. The authors also analyze the quantity-quality trade-off inherent in current data collection methods and suggest that this trade-off can be mitigated by redesigning incentives and task structures. This position proposes exploring new paradigms, such as the use of games, for sustainable data sourcing practices that maintain both quality and quantity in human-generated data for AI applications.

**Position:**

Yes

**Position In Title:**

Yes

**Related Work:**

4

**Strengths And Weaknesses:**

Strengths:

S1: The submission effectively highlights the importance of intrinsic motivations in engaging contributors for data collection, suggesting a more sustainable approach to data sourcing than traditional reliance on external incentives. This position is very meaningful because data collection is the foundation of the AI models.

S2: This submission offers a thorough exploration of the quantity-quality trade-off within current data collection systems.

S3: In this submission, the exploration of novel paradigms, such as using games for data collection, presents innovative and practical solutions that could enhance engagement and data quality.


Weaknesses:

W1: In this submission, the motivation of combining data collection systems with intrinsic may underestimate the challenges involved in designing and implementing such systems in different user groups and environments. Future scientific and technological progress requires professionals in their respective fields to accumulate their own data rather than a unified system, especially in the field of AI4S.

W2: In this submission, the proposed alternative data collection strategies, e.g. gamification, may lack sufficient empirical evidence or case studies to demonstrate their effectiveness compared to traditional methods.

W3: The success of the proposed system depends heavily on the continued participation of contributors, which may be difficult to achieve in practice.

**Support:**

2

---

> ### Author Rebuttal · Authors · 2025-03-31
>
> Thank you for your thoughtful review. We're glad the focus on intrinsic motivation and the quality-quantity trade-off came through clearly, and we appreciate your engagement with the idea of games as a new direction for data collection. Below, we respond to the concerns you raised.
>
>
> **W1: Underestimating the difficulty of designing intrinsically motivated systems**
>
> We strongly agree that designing systems grounded in intrinsic motivation is not trivial, and we discuss this challenge explicitly in Section 8. Intrinsically motivated data collection setups have to optimize for both data utility and contributor engagement, whereas traditional systems typically only focus on data utility – making intrinsically motivated systems inherently more challenging as the reviewer rightly suggests.
>
> However, in light of the current crisis in human data collection, our position is an attempt to get researchers unstuck by encouraging renewed attention to a dimension that has largely fallen out of today’s discussions. We are not suggesting a complete shift, but rather a gradual step-up toward designing systems that account for intrinsic drivers. While this is certainly challenging, it opens up a promising direction that, if successful, could enable the collection of millions of high-quality data points, as shown by systems like the ESP Game and ReCAPTCHA.
>
> We also want to clarify that we do not advocate for a single unified system. Instead, we argue that systems prioritizing contributor wellbeing and intrinsic motivation, even at smaller scales, are more likely to yield high-quality human data and create a virtuous cycle where engagement and quality reinforce one another. This stands in contrast to a vicious cycle where external incentives crowd out intrinsic motivation and require ever-increasing rewards to sustain high-quality participation.
>
> **W2: Lack of empirical evidence for gamification strategies**
>
> In Section 8, we discuss several real-world implementations, including “Games With A Purpose” such as the ESP Game, Peekaboom, and Verbosity, as well as more recent systems like QuickDraw, Iconary, and HumanOrNot. In fact, Reviewer av6X notes that this section is already rich with examples. That said, they also point out – and this may be your main concern as well – that specific gamification elements are not discussed in depth. We respond to that in detail in our response to av6X, but in short:
> - Going into detailed breakdowns digresses from the core argument, as our goal is to give ML researchers a conceptual footing to begin exploring this space, and
> - There is no one-size-fits-all answer – there are many gamification strategies, and the right approach depends heavily on context and requires iteration.
>
> **W3: Concerns about sustaining long-term participation**
>
> We agree that sustaining participation over time is a real challenge. Continued participation requires designing incentives that do not exhaust easily. While external incentives may work in the short term, they often lead to long-term issues such as degraded data quality and worker wellbeing. Intrinsic motivation, on the other hand, avoids this shortcut and instead confronts the harder design problem head-on. Supporting long-term engagement through intrinsic motivation requires careful incentive design that particularly tap into the fundamental human drives like curiosity, learning, and the desire to make sense of the world around them.
>
> In practice, designing incentives for continuous engagement has stumped game and product designers for decades and is far from uncharted territory, with many wildly successful games and products sustaining engagement over years. We believe similar design principles can inform how we build more sustainable human data collection systems moving forward.
>
> --
>
> Thank you again for highlighting the significance of this direction. We hope these changes would strengthen the paper’s case for rethinking how we sustainably and meaningfully engage human contributors in the future of AI. Please let us know if you have any other suggestions that you’d like us to consider.

---

### Decision · Program_Chairs · 2025-05-01

**Decision:**

Accept (poster)

**Comment:**

Reviewers agreed the topic is timely and worth discussing. Reviewer g2z8 noted the “direct actionable changes” suggested for both academic and industry stakeholders. Reviewer av6X supported the core argument but felt the pivot to games came too abruptly and would benefit from stronger integration with the earlier narrative development in the paper. Reviewer BHDJ challenged the idea that intrinsic motivation is always more effective. The authors responded that the issue is not financial rewards per se, but rather the risk of crowding out intrinsic drivers when extrinsic rewards dominate. In response, the authors suggest hybrid models, e.g. delayed compensation or “post-hoc recognition. Reviewer VBeJ remained unconvinced re: cultural generalizability, implementation challenges, etc.

Overall, while there are open questions around feasibility and scale, the paper stakes a clear position, addresses competing perspectives, and helps reframe an underexplored problem in ML practice.